# PROTOTYPICAL EVOLUATION FOR FEW-SHOT LEARNING IN VISION-LANGUAGE MODEL ADAPTATION

## ABSTRACT

Vision-Language Models (e.g., CLIP), with their immense capacity and extensive exposure to vast data during pre-training, have demonstrated a strong ability to capture real-world concepts. When fast adapted to downstream tasks with only a few labeled samples, parameter-efficient methods, such as prompt-based and adapter-based approaches, which adjust only a small portion of the parameters, have proven effective in reducing the escalating costs in large vision-language models. However, conventional efficient fine-tuning techniques, using task-specific objectives like cross-entropy loss, often lead to overfitting the downstream data distributions. This overfitting diminishes the model's ability to retain its original generalization capacity, especially on out-of-distribution (OOD) samples. Unlike the pretraining stage, where rich textual descriptions are available, fine-tuning is typically constrained to using only class names. This creates suboptimal text-image alignment in the shared feature space, as it may exacerbate image feature variance within the same class. To address this issue, we propose Prototypical Evolutionary Adaptation (PEA), leveraging off-the-shelf image centroids as prototypes to regulate image feature variance, mitigating the excessive feature variance within the same class caused by selective bias. Additionally, we introduce learnable shift vectors to capture the dynamics of class prototypes, ensuring that they remain compact and informative. Experiments across diverse datasets and model architectures in few-shot learning demonstrate that our approach consistently outperforms existing methods while maintaining robust generalization under varying distribution shifts.

## 1 INTRODUCTION

Vision-Language Models (VLMs) like CLIP have demonstrated impressive zero-shot classification abilities by learning a shared semantic space between visual and textual modalities. This success is driven by the model's ability to leverage vast datasets of web-scale image-text pairs during pre-training, allowing it to classify images into various categories using only prompts, such as "a photo of a [*class*]", without any additional training. While CLIP excels in these zero-shot tasks, its performance can be further enhanced in downstream tasks with limited labeled data. To address this, recent research has focused on developing parameter-efficient fine-tuning methods that reduce the number of trainable parameters while improving performance on few-shot learning tasks.

Parameter-efficient methods, such as prompt-based approaches like CoOp (Zhou et al., 2022c) and adapter-based (Gao et al., 2024) approaches like CLIP-Adapter, have made significant strides in adapting CLIP to few-shot learning tasks. These approaches introduce minimal additional parameters while achieving considerable performance improvements. However, despite their efficiency, these methods often suffer from overfitting on limited downstream data, particularly when relying solely on class names for fine-tuning, leading to a reduction in generalization performance, especially on out-of-distribution (OOD) samples(Kumar et al., 2022).

To address these limitations, we propose Prototypical Evolutionary Adaptation (PEA), a novel approach that builds upon the class prototype methodology. While conventional class-prototype methods such as Nearest Mean Classifier (NMC) use static prototypes based on feature averages, these prototypes can be biased and insufficient in capturing the true distribution of class features. Our method introduces dynamic prototypes that evolve throughout the fine-tuning process, leveraging

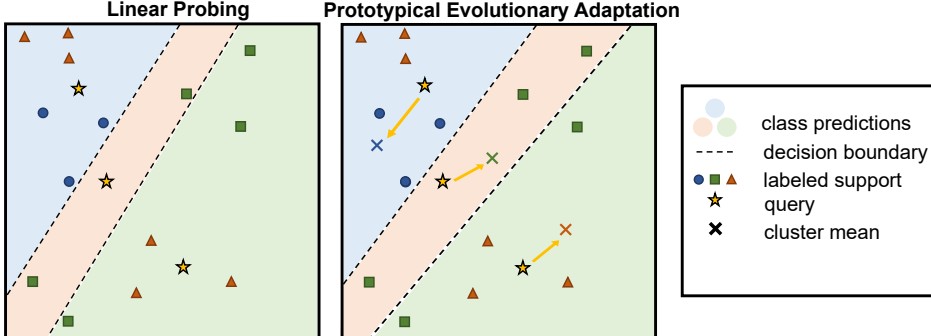

**Figure 1:** Overview of Prototypical Evolutionary Adaptation. The static class prototype within the visual feature space can be affected by selection bias, as well as the limited $K$ images per class. To address this, we propose PEA, which dynamically calibrates the biased prototypes during the learning process to ensure they are more accurate and informative.

learnable shift vectors that adjust the prototypes based on the underlying feature variance. This approach helps mitigate overfitting and enhances the representational capacity of the prototypes, ensuring they remain compact and informative across varying class distributions.

Moreover, we regulate the intra-class variance by leveraging off-the-shelf image centroids and adjusting them with learnable shift vectors, allowing PEA to better capture the diversity within each class. This calibration reduces the impact of biased prototypes that result from the limited availability of training samples in few-shot scenarios. By dynamically evolving these prototypes, PEA maintains the generalization power of the pre-trained model while improving alignment between the visual and textual modalities.

Extensive experiments across a variety of datasets and tasks demonstrate that PEA consistently outperforms existing few-shot learning methods, achieving robust generalization under distribution shifts. Our approach not only exceeds the performance of training-free methods but also provides comparable or better results than training-required methods, while maintaining efficiency in parameter usage. These results highlight the effectiveness of PEA as a powerful and scalable method for few-shot learning with VLMs.

## 2 RELATED WORKS

**Vision Language Models (VLMs)** In recent years, VLMs have attracted significant attention from researchers, emerging as a promising paradigm and have been successfully applied to numerous visual tasks. A notable example is CLIP (Radford et al., 2021), which leverages weak supervision by using the linguistic description of each image as a training signal. It underwent training on a vast corpus of 400 million web-crawled images and texts, achieving results competitive with supervised baseline. Then a crops of works (Goel et al., 2022; Li et al., 2022; Zhai et al., 2023) explored vision-language pretraining to obtain versatile applicable representations. Although, these pretrained VLMs have learned transferable representations for both vision and languages, adapting to downstream tasks remains a challenging research problem. There have been many tailored methods proposed to adapt VLMs for few-shot classification (Zhou et al., 2022c;a), semantic segmentation (Lin et al., 2023; He et al., 2023) and object dection (Mao et al., 2023; Wu et al., 2023).

**Efficient transfer leaning.** Given the large size of pre-trained VLMs like CLIP (Radford et al., 2021), efficiently fine-tuning these models for downstream tasks has become a central focus of recent research. The goal of parameter-efficient transfer learning is to achieve optimal performance with minimal modifications to the pre-trained model, which is particularly important in few-shot learning scenarios where labeled data is scarce. One prominent approach is prompt tuning, which optimizes only the input prompts while keeping the backbone of the model frozen. Methods like CoOp (Zhou et al., 2022c) introduced learnable textual prompts that adapt to downstream tasks

through back-propagation, allowing the model to leverage the rich knowledge embedded in the pre-trained weights. By tuning just the prompts, this approach minimizes the need to modify the model's core parameters, making it both efficient and effective for few-shot learning. However, despite strong performance gains, prompt tuning has been shown to face limitations in generalization, particularly when dealing with unseen classes. To address these challenges, CoCoOp (Zhou et al., 2022a) extends CoOp by incorporating visual features into the prompt generation process, enhancing the model's ability to generalize from base classes to novel ones. Another key strategy in parameter-efficient fine-tuning is adapter-based methods. Instead of fine-tuning the entire model, these methods introduce lightweight adapter modules that adjust the visual and textual representations of CLIP. CLIP-Adapter (Gao et al., 2024) refines the original vision and language embeddings by training task-specific adapters, which are inserted into pre-trained layers. This approach retains the efficiency of the model by limiting the number of trainable parameters while still improving task-specific performance. However, despite their efficiency, adapter-based methods still require additional computational cost during inference stage.

**Few-shot learning.** Few-shot learning approaches are typically divided into two main categories: metric-based methods and optimization-based methods. Metric-based methods aim to map samples into an embedding space where classification is performed based on the distance between the query samples and class prototypes. These methods rely on predefined, task-agnostic distance metrics to measure similarity between the samples and the class representatives. Commonly used metrics include cosine similarity, which calculates the cosine of the angle between two vectors in the embedding space, and Euclidean distance, which measures the straight-line distance between two points. One of the most well-known metric-based methods is Prototypical Networks (Snell et al., 2017), which computes a single prototype for each class and classifies new samples based on their proximity to these prototypes. While these methods are efficient, they may struggle to adapt to more complex tasks where a single prototype per class does not capture intra-class variations. Optimization-based methods, on the other hand, aim to learn optimal initial model parameters that can be quickly fine-tuned for new tasks using only a few labeled examples. Model-Agnostic Meta-Learning (MAML) (Finn et al., 2017) is a prominent example of this approach. In MAML, the model is trained to be sensitive to changes in task-specific data, allowing it to adapt rapidly with minimal updates. During the meta-training phase, MAML optimizes the model parameters on a set of base tasks so that it can quickly adapt to novel tasks with only a few gradient steps. In this paper, we utilize the limited supervision signal to better calibrate the biased mean estimation of frozen visual features rather than learning the metric.

## 3 PROBLEM SETTING AND PRELIMINARIES

Throughout the paper, we consider canonical image classification tasks using pre-trained VLMs (Radford et al., 2021; Goel et al., 2022; Zhai et al., 2023). Although our primary focus is on CLIP (Radford et al., 2021), it is important to highlight that the discussion could be extend to other VLMs, which shares similar characteristics.

**Problem setting.** Our objective is to efficiently fine-tune pre-trained vision-language models for various target downstream tasks, especially when only a limited number of examples are accessible for each category. Concretely, this problem can be denoted as a $N$-way $K$-shot classification task. In this context, the support set $S = \{(x_m, y_m)\}_{m=1}^{M=N \times K}$ consists of $N$ distinct classes, with $K$ labeled examples provided for each class, resulting in a total of $M$ samples.

**CLIP Zero-shot inference (Radford et al., 2021).** Classic CLIP is composed of of an image encoder $E_v$ and a text encoder $E_t$ parameterized by $\theta_v, \theta_t$ respectively. These encoders map the input into a shared $D$-dimensional representation space. Given the query image $x$ and a set of class names $C$, CLIP demonstrates the ability to predict the target label $y$ in a zero-shot manner. To achieve this, each class name is embedded within a manually tailored template to generate a prompt (*e.g.*, a photo of a [*class name*]). CLIP processes both the prompt and the query image to obtain a class-specific embedding $t_c = E_t(c)$ for each class and the sample embedding $u = E_v(x)$. Then we can compute the probability of assigning the query image into category $k$ using the dot product similarity, which is equivalent to cosine similarity, between the class embedding $t_k$ and the query

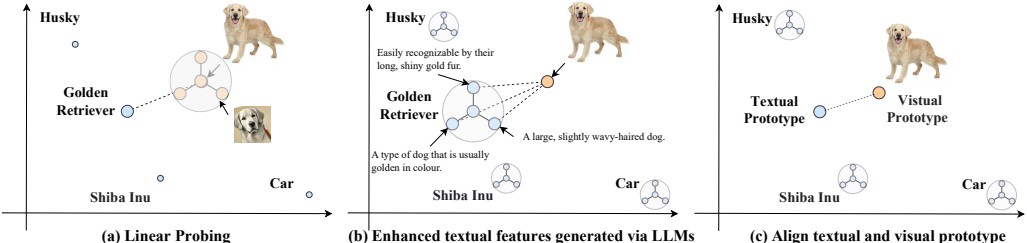

**Figure 2:** Motivation

image embedding $u$, normalized by a temperature factor $\tau$:

$$P(y = k \,|\, x) = \text{Softmax}(\langle t_k, u \rangle / \tau) = \frac{\exp(\langle t_k, u \rangle / \tau)}{\sum_{c=1}^{C} \exp(\langle t_c, u \rangle / \tau)}. \tag{1}$$

**Linear Probe (Wortsman et al., 2022) and Adaper (Gao et al., 2024).** One of the most straight-forward methods for adapting VLMs is Linear Probing (LP) (Radford et al., 2021; Wortsman et al., 2022). In this case, an additional linear layer $w \in \mathbb{R}^{D \times K}$ is appended to the top of the supported data sample embeddings. The goal is to learn a set of class-wise prototypes, $w_c$, that can generate softmax class scores for any given query visual embedding $u$.

$$\hat{P}(y = k \,|\, x) = \text{Softmax}(\langle w_k, u \rangle / \tau) = \frac{\exp(\langle w_k, u \rangle / \tau)}{\sum_{c=1}^{C} \exp(\langle w_c, u \rangle / \tau)}. \tag{2}$$

Formally, these class-specific prototypes, $w_c$, are optimized by minimizing the cross-entropy loss on the support samples, as shown in Equation 2. To enhance the generalization performance, inspired by Equation 1, the initialization of these learnable prototypes can be guided by CLIP's zero-shot prototypes, $t_k$, as also suggested in Wortsman et al. (2022); Kumar et al. (2022), which benefits the acceleration of convergence. Besides, it is worth noting that, in the absence of additional training, LP degenerates to zero-shot classification.

Furthermore, as mentioned in (Liang et al., 2022), the pre-training contrastive loss tends to maintain the *modality gap*, meaning that image and text embeddings occupy distinct regions in the shared embedding space. With this inherent gap, the mismatch objective between pre-training and LP will exacerbate the model's ability to generalize across various downstream tasks (Goyal et al., 2023). A simple rescue to this is Adaper (Gao et al., 2024; Zhang et al., 2022; Zhu et al., 2023), which trains a simple 2-layer bottleneck multilayer perception to output transformed sample embeddings instead of the original sample embeddings. Formally, given a hidden layer of dimension $H$, a ReLU activation function $\sigma$, and adapter weights $W_1 \in \mathbb{R}^{D \times H}$ and $W_2 \in \mathbb{R}^{H \times D}$, we compute the "adapted" embeddings as follows: $f(u) = W_2^T \sigma(W_1^T u)$. Adapters finally learn transformations to align sample embeddings to class embeddings. We can make a transformation to this formula as $\widetilde{u} = f(u)$ to fit Equation 2.

## 4 METHOD

In this section, we formally introduce our method PEA, where the overall pipeline is shown in Figure 1. Specifically, we start by explaining our motivation and then discuss how to evolve class prototypes. Finally, we present the complete algorithm.

### 4.1 MOTIVATION

In real-world scenarios, objects that share the same label can exhibit vastly different characteristics, as their appearances vary dramatically in terms of color, texture, shape, background, and style. These differences, ranging from subtle to significant, could be further amplified in the feature space after extraction by VLMs. As illustrated in Figure 2, the extracted visual features are highly diverse, and some have low similarity scores with their ground-truth class names. This rich visual diversity

challenges the effectiveness of simple prompt templates like 'a photo of a [*class*]', as such prompts may not sufficiently capture the detailed variations present in these images.

With the widespread use of GPT-3 (Brown, 2020) to generate descriptions, Menon & Vondrick (2023); Pratt et al. (2023); Roth et al. (2023) circumvented the challenges posed by rich visual diversity and leverage the knowledge embedded in Large Language Models(LLMs) for the automatic generation of class-specific descriptions. These descriptions aim to enhance the diversity of textual representations by focusing on the discriminative features of image categories, which are then aligned with the query images. However, in Figure 2, the detailed textual descriptions generated by LLMs may still exhibit low similarity scores with the features extracted from the query images. Furthermore, as pointed in Zhou et al. (2022b), even minor modifications to the prompt, *e.g.*, changing the prompt 'a photo of [*class*]' to 'a photo of a [*class*]', can give rise to a performance improvement of up to 6%. This sensitivity to specific wording suggests that overly detailed descriptions may actually degrade downstream performance due to the nuanced nature of language.

Since broad and generic class templates can be considered as the class centroids of detailed class-specific descriptions within the textual feature space, it is natural to extend this concept to the visual domain to account for rich visual diversity. A straightforward method to tackle this issue is to align the textual class prototypes with the visual class prototypes. However, this approach may suffer from selection bias in that the support dataset is randomly divided, and the informative class centroid is also affected by the number of shots ($K$-shot). Specifically, a larger $K$ leads to a more accurate estimation of the class centroid. Motivated by these challenges, we propose PEA to address the issues for accurate class centroid estimation.

## 4.2 PEA: PROTOTYPE EVOLUTIONARY ADAPTATION

To harness the powerful visual representations learned by large-scale pre-trained VLMs while overcoming the limitations of full adaptation and LP, class-prototype methods have been introduced. These methods extract features from the last layer of the pre-trained model and aggregate them to construct representative prototypes for each class. The most straightforward of these is the Nearest Mean Classifier (NMC) (Mensink et al., 2013), which computes a class prototype $\bar{c}_y$ for each class $y$ by averaging the feature representations of the supporting samples belonging to that class:

$$\bar{c}_y = \frac{1}{N \times K} \sum_{m=1}^{N \times K} \mathbb{1}(y = y_m) \cdot u_m, \tag{3}$$

where $\mathbb{1}(\cdot)$ denotes the indicator function. During inference, NMC assigns each test sample to the class whose class prototype is most similar to the sample's feature vector. This similarity is measured by either the smallest Euclidean distance (Janson et al., 2022) or the highest cosine similarity (Zhou et al., 2024) between the test sample's feature embedding and the class prototypes. Considering the dot product similarity measure, the predicted class label is obtained by:

$$\bar{y} = \arg\max_{y \in \{1, \cdots, C\}} \bar{P}(y \mid x), \quad \bar{P}(y \mid x) := \frac{\exp(\langle \bar{c}_y, u \rangle / \tau)}{\sum_{k=1}^{C} \exp(\langle \bar{c}_k, u \rangle / \tau)} \tag{4}$$

Throughout the entire few-shot learning process, we keep the CLIP model frozen and the classifier is implemented using class prototypes and can be represented by $N$ prototypes, *i.e.*, $W = [\bar{c}_1, \cdots, \bar{c}_N]$.

Though we already have a basic estimation of each class's mean centroid, this first-order moment during estimation lacks detailed statistical information about the true class distribution. Instead of indirectly altering feature embeddings through prompt tuning or image transformation to achieve unbiased estimations or capture higher-order statistical moments, we propose to directly introduce a learnable shift to the class prototypes to calibrate biased prototypes in its infancy. Since this adjustment dynamically calibrates the biased prototypes, resulting more informative class centroid. This is why we refer to it as Prototype Evolutionary Adaptation (PEA). The evolved prototype $\bar{c}'_y$ can then be notated by:

$$\bar{c}'_y = \bar{c}_y + \alpha \cdot \Delta_c \tag{5}$$

The hyperparameter $\alpha$ regulates the extent to which biased prototypes are adjusted during the evolution process. When $\alpha$ is small, the evolved prototypes remain close to the original biased prototypes,

**Table 1: Comparison to state-of-the-art methods on 11 classification tasks.** We report RN-50 CLIP model on 16-shot datasets. Prompt-learning and CALP methods results are directly extracted from Zhou et al. (2022c); Silva-Rodriguez et al. (2024). **Bold** denotes the highest results.

| Method | Pets | Flowers | FGVC | DTD | EuroSAT | Cars | Food | SUN | Caltech | UCF | ImageNet | Average |
|---|---|---|---|---|---|---|---|---|---|---|---|---|
| ZS | 85.77 | 66.14 | 17.28 | 42.32 | 37.56 | 55.61 | 77.31 | 58.52 | 86.29 | 61.46 | 58.18 | 58.77 |
| Rand LP | 71.63 | 92.73 | 34.63 | 60.60 | 73.38 | 69.20 | 66.92 | 63.07 | 87.55 | 70.94 | 52.24 | 67.54 |
| ZS-LP | 86.27 | 95.82 | 34.82 | 66.43 | 83.16 | 75.49 | 75.86 | 69.72 | 92.98 | 76.54 | 61.00 | 74.37 |
| CLAP | 88.51 | 94.21 | 33.59 | 66.41 | 80.07 | 75.12 | **78.55** | **70.78** | 91.93 | 76.29 | **65.02** | 74.57 |
| CoOp | 87.02 | 94.49 | 31.46 | 62.51 | 83.69 | 73.60 | 74.48 | 68.36 | 91.99 | 76.90 | 61.91 | 73.33 |
| PLOT | 87.21 | 94.67 | 31.49 | 65.60 | 82.23 | 72.80 | 77.09 | 69.96 | 92.24 | 77.26 | 63.01 | 73.94 |
| Tip-Adapter | 81.90 | 78.41 | 21.96 | 54.79 | 67.90 | 58.83 | 72.96 | 64.00 | 88.44 | 64.52 | 57.81 | 64.61 |
| APE | 87.98 | 91.96 | 31.23 | 67.38 | 78.40 | 70.45 | 78.37 | 69.59 | 92.29 | 74.49 | 63.43 | 73.23 |
| TaskRes | 86.28 | 95.82 | **34.82** | 66.45 | **83.15** | 75.48 | 75.86 | 69.72 | 93.00 | 76.54 | 61.01 | 74.38 |
| **PEA** | **88.99** | **96.06** | 33.90 | **68.50** | 78.56 | **75.90** | 77.42 | 69.73 | **93.35** | **79.41** | 64.88 | **75.15** |

preserving much of their initial characteristics. Conversely, a larger $\alpha$ value causes the evolved prototypes to incorporate more features from the base prototypes, effectively reducing the initial bias. Then the class-wise probabilities can be formulated as:

$$\bar{P}(y\,|\,x) := \frac{\exp(\langle \bar{c}'_y + t_y, u\rangle/\tau)}{\sum_{k=1}^{C}\exp(\langle \bar{c}'_k + t_k, u\rangle/\tau)} = \frac{\exp\left(\langle\, \bar{c}'_y, u\,\rangle/\tau + \langle\, t_y, u\,\rangle/\tau\right)}{\sum_{k=1}^{C}\exp(\langle \bar{c}'_k + t_k, u\rangle/\tau)} \quad (6)$$

**Connection to other parameter-efficient fine-tuning methods.** As discussed in (Kumar et al., 2022; Mukhoti et al., 2023), full fine-tuning can distort pretrained features and degrade performance, especially under mild distribution shifts. LP leverages the advantage of inheriting frozen pretrained features, achieving good performance under distribution shifts; however, it often results in unsatisfactory downstream performance.

A simple yet efficient remedy proposed in Wortsman et al. (2022); Ilharco et al. (2022); Kim et al. (2024) involves patching pretrained models by linearly interpolating weights between zero-shot models and fine-tuned models. This method implicitly edits the frozen representations in the *weight space*. Another line of work (Zhou et al., 2022c;a) aims to learn soft prompts by optimizing a continuous set of prompt vectors, which interferes with the frozen representations through the *input space*.

The most relevant works to ours are Yu et al. (2023); Sui et al. (2024), which steer the frozen features directly within the *embedding space*. Both methods focus on the textual feature space, and the experiments show that they yield only marginal improvements when applied to the visual feature space. In contrast, we exploit the intrinsic properties of the visual feature space. By only calibrating the biased prototypes, we further enhance few-shot learning with altering the frozen representations.

## 5 EXPERIMENTS

### 5.1 SETUP

**Datasets.** To evaluate the effectiveness of our few-shot learning approach, we conducted experiments on a diverse set of 11 publicly available image classification datasets, following the protocols established in prior works (Gao et al., 2024; Yu et al., 2023; Zhang et al., 2022). These datasets encompass a wide range of image recognition tasks: **Generic object recognition**: ImageNet (Deng et al., 2009), Caltech101 (Fei-Fei et al., 2004), **Fine-grained recognition**: Oxford Pets (Parkhi et al., 2012), Stanford Cars (Krause et al., 2013), Flowers102 (Nilsback & Zisserman, 2008), Food101 (Bossard et al., 2014), FGVCAircraft (Maji et al., 2013), **Satellite imagery classification**: EuroSAT (Helber et al., 2019), **Action recognition**: UCF101(Kay et al., 2017), **Texture classification**: DTD (Cimpoi et al., 2014), **Scene recognition**: SUN397 (Xiao et al., 2010). For the few-shot learning setup, we randomly selected $K$ examples per class, where $K \in \{1, 2, 4, 8, 16\}$, to fast finetune our models. We used the standard test sets provided with each dataset for evaluation, adhering to the same data splits as in previous studies (Yu et al., 2023; Zhou et al., 2022c). To assess the robustness of our methods to domain shifts, we performed out-of-distribution (OOD)

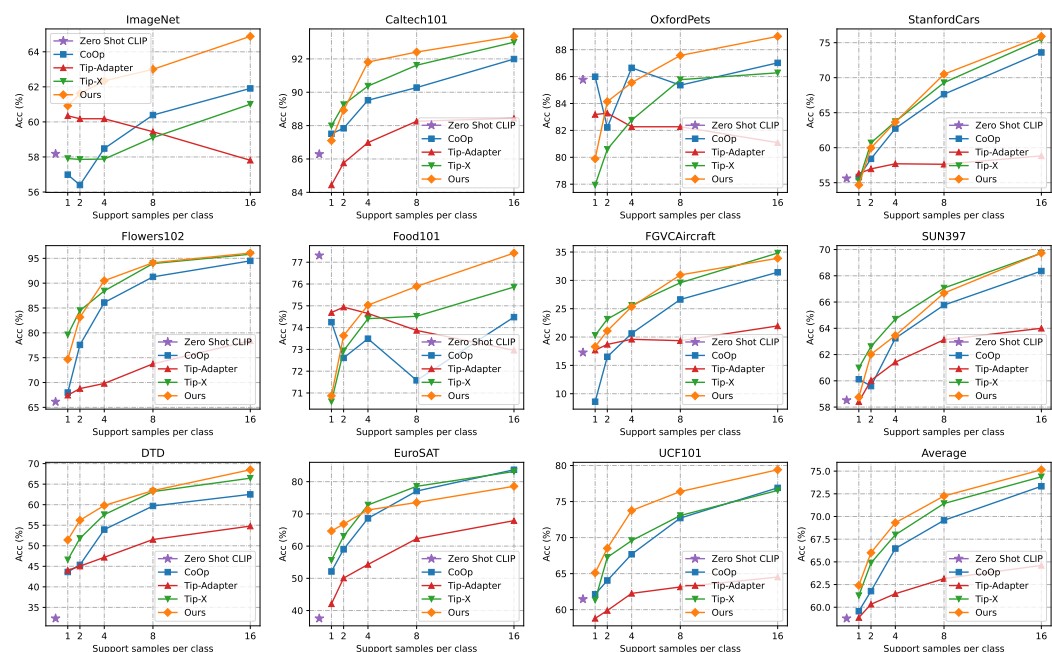

**Figure 3:** Results of few-shot classification on the 11 datasets. We evaluate the performance of our proposed method against different methods under 1, 2, 4, 8, and 16-shot settings.

**Table 2: Out-of-distribution generalization results.** 'Source' refers to in-distribution accuracy, while 'Target' represents out-of-distribution performance. All methods finetuned on 16 images per class from source dataset. **Bold** indicates best performance. Relative improvements are obtained for each methods with respect to zero-shot prediction.

| Method | Visual Backbone | Source Imagenet | Target | | | | |
|---|---|---|---|---|---|---|---|
| | | | -V2 | -Sketch | -A | -R | Avg. |
| Zero-Shot ICML'21 | ResNet-50 | 60.35 | 51.49 | 33.33 | 21.67 | 55.93 | 40.61 |
| Rand. Init LP ICML'21 | | $52.24_{(-8.11)}\downarrow$ | 41.85 | 15.93 | 10.72 | 29.95 | $24.61_{(-16.00)}\downarrow$ |
| CLIP-Adapter IJCV'23 | | $59.02_{(-1.33)}\downarrow$ | 48.15 | 14.63 | 15.75 | 46.29 | $31.21_{(-9.40)}\downarrow$ |
| TIP-Adapter ECCV'22 | | $57.81_{(-2.54)}\downarrow$ | 50.32 | 33.59 | 21.88 | 56.98 | $40.69_{(+0.08)}\uparrow$ |
| TaskRes(e) CVPR'23 | | $60.85_{(+0.50)}\uparrow$ | **56.47** | 32.80 | 19.90 | 55.93 | $41.28_{(+0.67)}\uparrow$ |
| ZS-LP CVPR'24 | | $61.00_{(+0.65)}\uparrow$ | 51.09 | 27.90 | 16.95 | 50.37 | $36.58_{(-4.03)}\downarrow$ |
| CLAP CVPR'24 | | $\mathbf{65.02}_{(+4.67)}\uparrow$ | 56.09 | 34.55 | 21.52 | **59.48** | $42.91_{(+2.30)}\uparrow$ |
| PEA | | $64.35_{(+4.00)}\uparrow$ | 56.26 | **36.34** | **23.07** | 61.34 | $\mathbf{44.25}_{(+3.64)}\uparrow$ |
| Zero-Shot ICML'21 | ViT-B/16 | 68.71 | 60.76 | 46.18 | 47.76 | 73.98 | 57.17 |
| Rand. Init LP ICML'21 | | $62.95_{(-5.76)}\downarrow$ | 52.48 | 29.22 | 29.40 | 50.54 | $40.41_{(-16.76)}\downarrow$ |
| CLIP-Adapter IJCV'23 | | $68.46_{(-0.25)}\downarrow$ | 59.55 | 39.88 | 38.83 | 64.62 | $50.72_{(-6.45)}\downarrow$ |
| TIP-Adapter ECCV'22 | | $53.81_{(-14.90)}\downarrow$ | 45.69 | 29.21 | 36.04 | 55.26 | $41.55_{(-15.62)}\downarrow$ |
| TaskRes(e) CVPR'23 | | $70.84_{(+2.13)}\uparrow$ | 62.15 | 43.76 | 43.91 | 71.59 | $55.35_{(-1.82)}\downarrow$ |
| ZS-LP CVPR'24 | | $69.73_{(+1.02)}\uparrow$ | 60.40 | 41.63 | 41.94 | 70.64 | $53.65_{(-3.52)}\downarrow$ |
| CLAP CVPR'24 | | $\mathbf{73.38}_{(+4.67)}\uparrow$ | 65.00 | 48.35 | 49.53 | 77.26 | $60.04_{(+2.87)}\uparrow$ |
| PEA | | $72.45_{(+3.74)}\uparrow$ | **65.32** | **49.48** | **51.37** | **78.05** | $\mathbf{61.01}_{(+3.84)}\uparrow$ |

experiments. Using ImageNet (Deng et al., 2009) as the source domain for adaptation, we evaluated our method on four of its variants as target domains: ImageNetV2 (Recht et al., 2019), ImageNet-Sketch (Wang et al., 2019), ImageNet-A (Hendrycks et al., 2021b), ImageNet-R (Hendrycks et al., 2021a). In this scenario, the model was trained using only a few labeled samples from the source domain (ImageNet), and the target datasets were exclusively used for testing. This setup allowed us to evaluate the model's domain generalization capabilities without any exposure to the target domains during training.

**Training details.** In our experiments, we leveraged pre-trained features from CLIP (Radford et al., 2021) using two primary backbone architectures: ResNet-50 (He et al., 2016) and ViT-B/16 (Doso-

vitskiy et al., 2021). The main experiments were conducted with both ResNet-50 and ViT-B/16, while the ablation studies specifically utilized ResNet-50 as the backbone. To make full use of the frozen features to accelerate the training process, so we extracted all the pre-trained features from the support sets and performed adaptation experiments based on these features. Following the methodology in Yu et al. (2023); Zhou et al. (2022b), we applied data augmentation during the feature extraction stage, including random zooms, crops, and flips. Each support sample was augmented 20 times to enhance the diversity of the training data. We employed the same text prompts for each dataset as specified in Yu et al. (2023); Zhou et al. (2022c). Training was carried out over 200 epochs using the SGD optimizer with a momentum of 0.9, inspired by the training strategies in Yu et al. (2023). We set the default initial learning rate to $2 \times 10^{-3}$ to prevent underfitting on the support sets. The learning rate was scheduled to decrease during training following a cosine decay pattern. All experiments are conducted on a single NVIDIA GeForce RTX 4090. To ensure robustness, all experiments were run with three different random seeds, and the results were averaged across these runs. Our method introduces a calibration strength parameter, denoted as $\alpha$, which adjusts the influence of the prototypes during adaptation. By default, $\alpha$ is set to 0.5 for all datasets, providing a balance between the biased and evolved prototypes. We also explored the impact of varying $\alpha$ values in our ablation studies.

**Baselines.** To evaluate the effectiveness of our proposed method, we compare it against several baseline approaches, which we organize into four distinct groups based on their methodologies and how they interact with pre-trained models. (1) **Zero-shot and random LP** (Radford et al., 2021): This group serves as a basic benchmark. It includes the zero-shot CLIP model, which uses prompts like "a photo of a [*class*]" without any additional training. Additionally, a linear classifier with random initialization is trained on top of the frozen pre-trained CLIP visual encoder's features. (2) **Improved LP Methods** (Wortsman et al., 2022; Silva-Rodriguez et al., 2024): These methods enhance standard linear probing by leveraging prior knowledge from textual embeddings. Classifier weights are initialized using class name prototypes derived from textual features, providing a better starting point for learning. They also introduce additional constraint terms during training to more effectively capture class-specific characteristics. (3) **Prompt Tuning Methods** (Implicit Representation Editing via Input Space) (Zhou et al., 2022c; Chen et al., 2023): Techniques like Context Optimization (CoOp) learn continuous prompt vectors through back-propagation. (4) **Methods Directly Altering the Feature Space** (Yu et al., 2023): This group includes approaches like TaskRes, which directly steers the frozen features in the textual embedding space using a task-specific residual connection.

## 5.2 Results

**Few-shot results.** We compare our proposed method, PEA, with several baseline methods in the few-shot learning setting, as summarized in Table 1. Across 12 datasets, PEA consistently demonstrates superior performance, achieving the highest average accuracy of 75.15%. Notably, it excels on datasets such as Oxford Pets (88.99%), Flowers102 (96.06%), and UCF101 (79.41%). Furthermore, as shown in Figure 3, we observe that as the number of images per class increases, the more informative class centroids lead to significant performance improvements.

## 6 Conclusion

In this paper, we revisit classic prototype-based methods in Vision-Language Models (VLMs) and propose a novel approach called Prototypical Evolutionary Adaptation (PEA). PEA refines the process of obtaining accurate class prototypes within the visual feature space by dynamically calibrating them throughout the fine-tuning process. This accurate class propotype will benefit the linear probing in the context of few-shot leanring. We conduct extensive experiments to evaluate the effectiveness of PEA on CLIP few-shot classification tasks and out-of-distribution generalization. Our method consistently outperforms state-of-the-art adapter-based and prompt-based approaches, demonstrating its superior performance. In future work, we aim to explore the application of PEA in other tasks and scenarios, such as test-time adaptation.

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
