# OpenReview forum: "Prototypical evoluation for few-shot learning in vision-language model adaptation"
_ICLR.cc/2025/Conference — ICLR 2025 Conference Withdrawn Submission_

### Official Review · Reviewer_BjJH · 2024-10-31

**Soundness:** 2
**Presentation:** 1
**Contribution:** 1
**Rating:** 1
**Confidence:** 5

**Summary:**

This paper introduces Prototypical Evolutionary Adaptation (PEA) to enhance few-shot adaptation in Vision-Language Models (VLMs), particularly CLIP. The authors propose learnable shift vectors for visual prototypes, which capture few-shot knowledge to improve zero-shot CLIP performance on downstream datasets. Experimental results across 11 few-shot learning datasets and 4 ImageNet variant datasets validate the effectiveness of PEA in adapting VLMs to downstream tasks.

**Strengths:**

1. The structure of this paper is clear.
2. The research question—few-shot adaptation of VLMs—is both well-established and of practical significance.
3. Experimental results across 15 datasets show that the proposed method outperforms selected baselines in few-shot adaptation and domain generalization tasks.

**Weaknesses:**

1. The concept of introducing learnable shift vectors has already been explored and demonstrated by TaskRes [1] and TPE [2], as noted by the authors. Although they claim these methods are effective only in the text space, other works, such as Tip-Adapter [3], ProtoCLIP [4] and APE [5], have also proposed approaches to modulate visual features in VLM adaptation. Therefore, I find that this paper lacks novelty in its approach and contributions.
2. The experiments are insufficient to fully support the authors' claims. For instance, in Line 85, the authors state that the proposed method is efficient in parameter usage, yet no supporting experiments are provided. Additionally, there is no evidence demonstrating how the proposed method regulates intra-class variance or how this regulation contributes to performance improvements.
3. The results of other baseline methods in Tables 1 and 2 appear unreliable, showing large discrepancies from their respective original papers. For example, Tip-Adapter-F’s 16-shot performance on ImageNet is reported as 65.51% in its original paper [3], yet in this paper, it is reported as 57.81% (Table 1), which is even lower than the zero-shot CLIP performance. Similarly, the reported performance of other baselines, including APE, TaskRes, and CoOp, is also notably lower than the results in their original papers.
4. The proposed method is not clearly explained. How is the learnable shift vector updated? From my understanding, it should also be updated using cross-entropy loss on the few-shot sample set. However, in the abstract, the authors state that '*using task-specific objectives like cross-entropy loss often leads to overfitting on downstream data distributions*,' which seems contradictory.
5. The paper appears largely incomplete. For instance, there are no ablation experiments to validate the effectiveness of the proposed components, and there is even no accompanying text to describe the results in Table 2.

**Minor issues**:
1. Typo in the title: Evoluation -> Evolution. The authors are suggested to proofread the paper before submitting it to a rigorous conference.
2. Figure 1 is not referenced in the main text, and its purpose is unclear.
3. The caption for Figure 2 is incomplete.
4. What the dashed lines represent in Table 1?
5. $\Delta_c$ in Equation 5 is not defined.


[1] Yu, T., Lu, Z., Jin, X., Chen, Z., & Wang, X. (2023). Task residual for tuning vision-language models. In *Proceedings of the IEEE/CVF Conference on Computer Vision and Pattern Recognition* (pp. 10899-10909).

[2] Sui, E., Wang, X., & Yeung-Levy, S. (2024). Just Shift It: Test-Time Prototype Shifting for Zero-Shot Generalization with Vision-Language Models. *arXiv preprint arXiv:2403.12952*.

[3] Zhang, R., Zhang, W., Fang, R., Gao, P., Li, K., Dai, J., ... & Li, H. (2022). Tip-Adapter: Training-Free Adaptation of CLIP for Few-Shot Classification. In *European Conference on Computer Vision* (pp. 493-510).

[4] Palanisamy, K., Chao, Y. W., Du, X., & Xiang, Y. (2023). Proto-clip: Vision-language prototypical network for few-shot learning. *arXiv preprint arXiv:2307.03073*.

[5] Zhu, X., Zhang, R., He, B., Zhou, A., Wang, D., Zhao, B., & Gao, P. (2023). Not all features matter: Enhancing few-shot clip with adaptive prior refinement. In *Proceedings of the IEEE/CVF International Conference on Computer Vision* (pp. 2605-2615).

**Questions:**

1. Why is the performance reported in this paper significantly different from that in the original paper? How were these results obtained? Did you attempt to reproduce their results but possibly implement it incorrectly?
2. What are the key novelties of this paper, compared to [1]-[5] listed in the weaknesses section?

---

### Official Review · Reviewer_4nCD · 2024-11-03

**Soundness:** 2
**Presentation:** 1
**Contribution:** 2
**Rating:** 3
**Confidence:** 4

**Summary:**

This work introduces a method called Prototypical Evolutionary Adaptation (PEA) to improve few-shot learning in VLMs by dynamically adjusting class prototypes with learnable shift vectors. PEA aims to reduce intra-class feature variance, enhancing model generalization across various datasets and distribution shifts.

**Strengths:**

PEA achieved good generalization across diverse datasets, outperforming existing few-shot adaptation techniques

**Weaknesses:**

1. The manuscript seems not completed (e.g. insufficient results and discussion, no ablation study).
2. The motivation is not convincing
- Why dynamic prototype evolution is essential compared to alternative methods. For example, many few-shot learning methods use static prototypes with effective regularization to control variance. It is not fully explained why these existing methods are insufficient.
- The potential advantages of introducing learnable shift vector over traditional static prototypes in feature space were not analyzed, making the introduction of learnable shift vector somewhat abrupt.
3. The method is not properly introduced
- Inadequate description of prototype update mechanism: the authors emphased that PEA introduces the use of a learnable shift vector but does not fully explain how these shift vectors are initialized, updated and regularized.
- Too much content about other works in the method section, hard to recognize which part is newly proposed.
- Hyperparamether α in eq.5 lacks detailed discussion or ablation study on how α affects model generalization
4. The evavlation is not comprehensive (e.g. How PEA behaves across different types of distribution shifts?)

**Questions:**

1. Figure 2 is confusing, can you add captions to describe it?
2. The notations in equations were not properly defined.
3. Can the authors add some visualization to show the effectiveness of PEA
4. Does PEA require significantly more computational cost?
5. Table 2 was not quoted in the main text.
6. "We also explored the impact of varying α values in our ablation studies", there is no related content.

---

### Official Review · Reviewer_GSCP · 2024-11-03

**Soundness:** 2
**Presentation:** 1
**Contribution:** 2
**Rating:** 3
**Confidence:** 4

**Summary:**

The authors proposed a slight variation of the standard prototype-based inference in the CLIP model, by keeping the backbone frozen and adjusting the class-prototype evolutionary. The proposed method, PEA, is then tested against the current few-shot classification benchmark and compared with SOTA models. Results show the superiority of PEa compared to other models in the base benchmark.

**Strengths:**

The proposed method seems a very conceptually simple, computationally cheap, and easy-to-use, approach to few-shot classification that preserves all the knowledge of massive pre-training (like in the CLIP model) during training.

**Weaknesses:**

In section "4.1 Motivation" the authors discuss limitations related to the text-handling side of CLIP-based classification. For instance, they discuss the limitations of text prompt engineering and refer to textual class prototypes as a defective method. However, I don't see how they are solving the issue here. The authors said that PEA addresses the issues for accurate class centroid estimation, but the proposed method is not dedicated to text embeddings, and the accurate centroid is for images only. Furthermore, the issue related to the limitations in text prompt engineering has been tackled by the Zhou et al (2022b) study (the CoOp model) as the authors also clarify. I don't see the connection between the text engineering issue and the PEA solution.

I struggled a lot with understanding the rationale behind eq. (6). I'm not saying it's problematic, because the performance is pretty interesting, but I didn't get how the authors arrived at that mathematical conclusion. First of all, what is $\Delta_c$ in eq. (5)? And the evolution proposed in the same equation starts from a base class-prototype and then modifies it. But how? If the class-prototype is fixed and computed once (it's just the average of image embeddings of the examples belonging to a class in the training set), what is the additional, iterative, information that can be obtained through the training process to perform this evolutionary step? I think it is fundamental to understand this step because this brings directly to eq. (6), where the general logic of class-propotype needs to be applied in the context of a contrastive loss. Here it is proposed that the image class-prototype can be added to the text embedding. But why? What's the mathematical justification for that? There are no other solutions? Is this based on some theoretical results not reported in the current manuscript? I think section 4.2 needs a huge refactoring, with more in-depth mathematical considerations and explanations.

A related question is, and I highlight that this is the main concept of all the paper in my opinion, what is a BIASED prototype? The parameter $\alpha$ is intended to reduce the initial bias. This is a core concept that needs to be explained extensively. I really encourage the authors to dedicate a whole section to this concept, with toy experiments and simulations, to clarify what is a biased prototype and why PEA is adding real value in fixing the bias problem. I don't think Fig. 1 mentioning the adjustment of this bias is enough.

In line 393, "ablation studies" is mentioned, but there is no reference. I think there is the need, if not done yet, to test the prototype approach without the evolutionary step, which is crucial to understand whethe the base prototype kept fixed and inserted in eq. (6) is not enough.

It's not clear which text prompt authors use at the end to perform the experiments. If "a photo of a []" has been used it should be put maybe in the training details.

In the model comparison, at least two popular models are missing, that is, CoCoOp (https://arxiv.org/abs/2203.05557), and MaPLe (https://arxiv.org/abs/2210.03117). But I please the authors to carefully check in the literature more relevant and recent models adapting CLIP for few-shot classification to ensure the model comparison is up-to-date.

In eq. (1), (2), (4), and (6), the term $x$ keeps appearing on the left-hand side of the equation as a conditioning variable $P(y|x)$ but never appears in the right-hand side equation.

"Evoluation" in the title

**Questions:**

Did the authors try several types of text prompts to see if PEA is robust to text prompt variations?

---

### Official Review · Reviewer_q7v7 · 2024-11-04

**Soundness:** 2
**Presentation:** 1
**Contribution:** 1
**Rating:** 3
**Confidence:** 4

**Summary:**

The paper finds that applying a simple learnable shift during the fine-tuning of CLIP in few-shot scenarios yields promising performance. However, it lacks sufficient motivation, and the overall presentation appears underdeveloped, giving an impression of incompleteness. The experimental results are marginal, and the analysis fails to provide insights into what makes this methodology particularly unique or meaningful.

**Strengths:**

- Fine-tuning vision-language foundation models (e.g., CLIP) for downstream tasks is increasingly crucial and practical in today’s research landscape.
- The proposed methodology is simple yet effective, demonstrating state-of-the-art performance.

**Weaknesses:**

**Paper Presentation and Completeness:** The paper appears incomplete, and the overall presentation is lacking. It is unclear what Figure 1 aims to convey, and Figure 2 lacks supporting experiments. While an ablation study is mentioned in Line 393, it does not actually exist, and the Experiment Results section in Line 412 only covers the few-shot setting, with no explanation provided for Table 2. Additionally, Figure 3 does not showcase the high-performing models from Table 1.

**Motivation and Supporting Examples:** In the Motivation section, the authors claim that methods using large language models (LLMs) are ineffective in enhancing performance, yet no supporting experiments are provided. Many studies report that LLM-based textual enrichment contributes to performance improvements, so countering these claims would require experimental evidence.

**Clarification of Technical Novelty and Differentiation:** I understand the main difference between the proposed methodology and TaskRes is that TaskRes applies a learnable shift to the textual features, whereas this approach applies it to the image features. However, it remains unclear how this approach differs from adapter-based methods. Specifically, how does directly updating the image features differ from applying a learnable shift? A detailed comparison with Tip-Adapter would be helpful, along with illustrative figures in future papers. Additionally, isn't Tip-Adapter set up to train only the first layer?

Additionally, the performance improvement appears marginal, with some datasets showing lower results compared to existing baselines. Given this, what are the core strengths of the proposed approach?

**Related Work and Additional Comparisons:** Selection bias resulting from limited images per class could potentially be mitigated by generating synthetic datasets using recent text-to-image generation models. Several studies [1, 2, 3] have shown promising results in similar settings (fine-tuning CLIP with few-shot or zero-shot datasets). Thus, it would be beneficial to compare and discuss the proposed methods with synthetic dataset generation approaches.

[1] He et al. "Is synthetic data from generative models ready for image recognition?" ICLR. 2023.
[2] Zhang et al. "Prompt, generate, then cache: Cascade of foundation models makes strong few-shot learners." CVPR. 2023.
[3] Udandarao et al. "Sus-x: Training-free name-only transfer of vision-language models." ICCV. 2023.

**Relevance of Out-of-Distribution (OOD) Comparisons:** In Table 2, the listed methods are not designed for out-of-distribution (OOD) tasks, so it's unclear why they are compared with OOD studies. It would be more appropriate to include comparisons with OOD-specific studies like WiSE-FT.

**Questions:**

- Is the parameter $\alpha$ fixed across all datasets, similar to CLAP?
- Minor comment: There is a typo in the paper's title (evoluation -> evolution).

---

### Note · Authors · 2024-11-13

I have read and agree with the venue's withdrawal policy on behalf of myself and my co-authors.